# Thermoelectric Properties of N-Type Poly (Ether Ether Ketone)/Carbon Nanofiber Melt-Processed Composites

**DOI:** 10.3390/polym14224803

**Published:** 2022-11-08

**Authors:** Antonio Jose Paleo, Beate Krause, Delfim Soares, Manuel Melle-Franco, Enrique Muñoz, Petra Pötschke, Ana Maria Rocha

**Affiliations:** 12C2T-Centre for Textile Science and Technology, Campus Azurem, University of Minho, 4800-058 Guimarães, Portugal; 2Leibniz-Institut für Polymerforschung Dresden e.V. (IPF), Hohe Str. 6, 01069 Dresden, Germany; 3CMEMS Research Center, Campus Azurem, University of Minho, 4800-058 Guimarães, Portugal; 4CICECO—Aveiro Institute of Materials, Department of Chemistry, University of Aveiro, 3810-193 Aveiro, Portugal; 5Facultad de Física, Pontificia Universidad Católica de Chile, Santiago 7820436, Chile

**Keywords:** carbon nanofibers, poly(ether ether ketone), conductive polymer composites, Seebeck coefficient, variable-range hopping, electronic doping

## Abstract

The thermoelectric properties, at temperatures from 30 °C to 100 °C, of melt-processed poly(ether ether ketone) (PEEK) composites prepared with 10 wt.% of carbon nanofibers (CNFs) are discussed in this work. At 30 °C, the PEEK/CNF composites show an electrical conductivity (σ) of ~27 S m^−1^ and a Seebeck coefficient (S) of −3.4 μV K^−1^, which means that their majority charge carriers are electrons. The origin of this negative Seebeck is deduced because of the impurities present in the as-received CNFs, which may cause sharply varying and localized states at approximately 0.086 eV above the Fermi energy level (E_F_) of CNFs. Moreover, the lower S, in absolute value, found in PEEK/CNF composites, when compared with the S of as-received CNFs (−5.3 μV K^−1^), is attributed to a slight electron withdrawing from the external layers of CNFs by the PEEK matrix. At temperatures from 30 °C to 100 °C, the σ (T) of PEEK/CNF composites, in contrast to the σ (T) of as-received CNFs, shows a negative temperature effect, understood through the 3D variable-range hopping (VRH) model, as a thermally activated hopping mechanism across a random network of potential wells. Moreover, their nonlinear S (T) follows the same behavior reported before for polypropylene composites melt-processed with similar CNFs at the same interval of temperatures.

## 1. Introduction

The electrical features of conductive polymer composites (CPCs) based on insulating polymers and carbonaceous materials have been extensively investigated since the end of the last century [1,2,3,4]. This is because the utilization of CPCs ranges from the high conductivities required for electromagnetic radiation shielding, electromagnetic interference (EMI), self-regulating heaters, and self-sensing materials [5] to the low conductivity of high dc-voltage cables and spacecraft components [6]. In this respect, carbon nanofibers (CNFs) produced by the chemical vapour deposition (CVD) of catalyst nanoparticles under gaseous hydrocarbons become very attractive among carbonaceous particles due to the combination of their interesting levels of conductivity (σ ~ 10^4^ S m^−1^) and low-cost processes [7]. In morphologic terms, the CNFs produced by CVD exhibit tubular hollow cores surrounded by two or more outer layers of different ordered structure, with total diameters and lengths ranging from 50 to 200 nm and 50 to 100 µm, respectively [8,9]. On the other hand, it is convenient to establish direct relationships between the intrinsic electronic properties of carbonaceous materials, such as the electrical conductivity (σ) and Seebeck coefficient (S), and the resulting CPCs derived from them to optimize the final properties of the latter ones. In this light, the Seebeck coefficient (S), which shows the voltage generated by a semiconductor when subjected to a temperature difference, enables knowing the majority carrier type present in that semiconductor [10]. In short, n-type semiconductors have negative S and a majority of electrons, while p-type semiconductors have positive S and a majority of holes [11]. It should be remembered that the dominant charge carrier is an important property in semiconductors because it defines their ultimate use as components of energy-harvesting devices such as solar and thermoelectric cells [12]. For instance, thermoelectric generators (TEGs) typically consist of pairs of p- and n-type materials, while a solar cell is composed of p and n-type materials joined together, which is called the p-n junction.

It is in this respect that different contents of n-type CNFs are used in this work to prepare melt-processed poly(ether ether ketone) (PEEK) composites. The aim was to endow this insulating polymer matrix with electrical functionality and to achieve air stable n-type thermoelectric composites without the use of any type of complex doping strategy. PEEK is a thermoplastic polymer characterized by high melting and glass transition temperatures that presents excellent specific strength and corrosion resistance [13]. After preparing conductive PEEK/CNF composites using a simple, cost-effective and scalable melt-mixing procedure, a detailed analysis is performed by comparing the electronic properties (σ and S) of the as-received CNFs and the PEEK composites prepared with 10 wt.% of those CNFs. It is shown that the CNFs transfer their n-type character to the PEEK/10 wt.% CNF composites. Moreover, the PEEK/10 wt.% CNF composites showed lower σ and S (in absolute value) than the CNFs. In particular, the role of PEEK on the Seebeck coefficient of CNFs is examined by implementing a chemical model that measures the charge transfer existing between the outer graphitic shells of CNFs in contact with the PEEK chains [14]. Interestingly, a similar effect (slight p-type doping of carbon nanostructure by the PEEK matrix) has been observed in PEEK composites melt-processed with commercial multiwall carbon nanotubes (MWCNTs) Nanocyl^TM^ NC7000 [15,16] and single-wall carbon nanotubes (SWCNTs) Tuball^TM^ [17]. In addition, quite unexpectedly, the electrical conductivity σ (T) of PEEK/10 wt.% CNF composites does not follow the σ (T) found in CNFs. That is, while the σ (T) of CNFs presents a reduction in electrical conductivity with temperature (positive temperature effect), the σ (T) of PEEK/10 wt.% CNF composites shows an increase in electrical conductivity with temperature (negative temperature effect). Finally, the cause of their n-type character is analysed through the modelling of S between 30 °C and 100 °C. All these outcomes are discussed in order to correlate the electronic properties of these PEEK/10 wt.% CNF composites processed with a conventional methodology with their potential use, a necessary practice for optimizing applications, which need of CPCs as building blocks. To our knowledge, the σ (T), S (T) analysis, and corresponding modelling presented here has not been discussed before for this particular type of CPCs.

## 2. Materials and Methods

### 2.1. Materials

A poly(ether ether ketone) powder VESTAKEEP^®^ 1000P (Evonik Industries AG, Essen, Germany) with a melt volume flow rate of 150 cm^3^·10 min^−1^ (380 °C, 5 kg) and a density of 1300 kg·m^−3^ was used as polymer matrix. Carbon nanofibers (Pyrograf^®^ III PR 24 LHT XT) supplied by ASI, Cedarville, OH, USA with bulk densities in the range of 0.016–0.048 g·cm^−3^ and lengths of 30–100 µm were chosen for processing melt-mixed PEEK-based composites with thermoelectrical properties. Pyrograf^®^ III CNFs’ properties have been extensively investigated [18,19,20]. The CNF grade used in this study is grown at 1100 °C by CVD and thermally treated at 1500 °C in inert atmosphere after processing. At the end, the as-received CNFs show a hollow cylinder surrounded by an outer double-wall structure, as shown in Figure 1b. 

### 2.2. Material Processing

The PEEK/CNF composites were melt mixed in a small-scale conical twin-screw microcompounder (Xplore Instruments BV, Sittard, The Netherlands) having a volume of 15 cm^3^. The polymer composites were mixed at 360 °C for 5 min with a rotational speed of 250 rpm. The CNFs and the polymer in powder form were alternatively filled into the main hopper of the compounder. The melt-mixed PEEK/CNF composite strands were then pelletized and compression-molded at 360 °C for 1 min with a hot press PW40EH (2 min preheating, max. force 50 kN, 0.5 min cooling in minichiller, polyimide foil as separation foil) to plates with diameter of 60 mm and thickness of ca. 0.3 mm. At the end, PEEK/CNF composites with CNF concentrations of 5, 7.5 and 10 wt.% were produced.

### 2.3. Morphological Analysis

The as-received CNFs were observed using ultra-high-resolution field emission gun scanning electron microscopy (FEG-SEM, NOVA 200 Nano SEM, FEI Company, Hillsboro, OR, USA) and a transmission electron microscope (TEM, JEOL JEM-2100) operating a LaB6 electron gun at 80 kV. TEM images were acquired with a “One-View” 4k × 4k CCD camera at minimal under-focus to achieve visibility of the CNF surface layers. Compression-moulded plates of the PEEK/CNF composite were cryo-fractured in liquid nitrogen and the surface was covered with 3 nm platinum before examining using a scanning electron microscope (SEM) (Ultra plus microscope, Carl Zeiss GmbH, Germany, field emission cathode) at 3 kV.

### 2.4. FTIR and DSC Analysis

Infrared measurements (FTIR, IRAffinity-1S, Shimadzu, Kyoto, Japan) were performed at room temperature in transmission mode from 600 to 2000 cm^−1^. FTIR spectra were collected with 40 scans and a resolution of 4 cm^−1^ at room temperature.

Differential scanning calorimetry measurements (DSC) were performed in argon atmosphere using a DSC Q20 instrument (TA Instruments, New Castle, DE, USA). The specimens were heated from 40 °C to 380 °C at a rate of 10 °C min^−1^ in nonisothermal experiments to eliminate any previous thermal history and then cooled down to 40 °C at a rate of 10 °C min^−1^. Following this analysis, the samples were heated to 380 °C at the same rate of 10 °C min^−1^.

### 2.5. Thermolectric Analysis

The Seebeck coefficient and volume resistivity of the PEEK/CNF composites and CNF powder were determined using the self-constructed equipment TEG at Leibniz-IPF [21,22]. A PVDF tube (inner diameter 3.8 mm, length 16 mm) filled with the CNF powder and closed with copper plugs was used for the thermoelectric measurements of the CNFs [16]. The PEEK/CNF samples were cut in strips of 15 mm × 4.5 mm and painted with conductive silver ink at their ends before testing. The thermovoltage and electrical resistance were performed using the Keithley multimeter DMM2001 (Keithley Instruments, Cleveland, OH, USA) with a free insert length of 12 mm between the two copper electrodes. The volume resistivity was measured using a 4-wire technique. The conductivity values represent the arithmetic means of ten measurements on two strips. The measurements of S were performed on the same strips at the mean temperatures of 30 °C (303.15 K), 40 °C, 60 °C, 80 °C, and 100 °C (373.15 K) by applying temperature differences between the two copper electrodes up to ±8 K around the mean temperature in 2 K steps. The Seebeck coefficient at each temperature was calculated as the average of eight thermoelectric voltage measurements. This measurement was repeated 3 times, and the means are reported. The samples containing 5 and 7.5 wt.% CNF were not conductive enough to perform measurements of the Seebeck coefficients. Thus, Seebeck coefficients are not reported for them. In particular, their electrical volume conductivities were 8 × 10^15^ Ohm·cm and 2 × 10^7^ Ohm·cm, respectively (measured with a Keithley 8009 Resistivity Test Fixture combined with electrometer Keithley E6517A). 

### 2.6. Thermal Conductivity

The thermal conductivity (*k*) of the PEEK/10 wt.% CNF composites was calculated from the product of thermal diffusivity, density, and specific heat capacity. The thermal diffusivity was measured on circular samples (diameter 12.5 mm, thickness 1.8 mm) through the plate thickness using the light flash apparatus LFA 447 NanoFlash (Netzsch-Gerätebau GmbH, Selb, Germany) at 30 °C (303.15 K), 40 °C, 60 °C, 80 °C, and 100 °C (373.15 K). The specific heat capacity was calculated by comparing the signal heights between the PEEK/10 wt.% CNF composite and the reference Pyroceram 9606 (with known specific heat capacity) using the LFA 447 NanoFlash software. The density of the PEEK/10 wt.% CNF composite was determined using the buoyancy method. The given values represent the means of four measurements.

### 2.7. Electronic Charge Transfer Modeling

The contact charge transfer between the PEEK chains and the external graphitic shells of CNFs outlined by the hexagonal graphene flake shown in Figure 2 was computed following on previous studies [14,23]. For this, the molecular geometry and the charge transfer of PEEK oligomers with up to three monomers adding up to a total length of 4.6 nm adsorbed on a graphene flake of 5.6 nm diameter and 912 atoms were optimized. The calculations were performed with the GFN1-xTB (G: geometries; F: frequencies; N: noncovalent interaction) tight-binding model, which allows for computing systems with thousands of atoms [24], while the charge transfer was computed by adding up the CM5 partial charges produced by the same Hamiltonian [25] on the PEEK oligomer and hexagonal graphene flake.

## 3. Results and Discussion

### 3.1. Morphological Analysis 

Figure 1 shows representative SEM and TEM images of CNFs. As shown in Figure 1b, the CNF diameter is of ~90 nm, which matches well with the mean diameters measured in previous works for the same type of CNFs [14,20]. The single CNF presents a straight and cylindrical hollow tube of around 42 nm with a surrounding double structure (Figure 1b). The inner part of the double layer shows an arrangement of compacted graphitic layers with a total thickness of ~7.4 nm. The outer part of the double layer shows a larger size of ~15.8 nm, and it is composed of graphitic sheets, though their morphology is not as perfect and compacted as that observed in the inner layer [14,23]. Figure 3 shows representative SEM images of the PEEK/CNF 10 wt.% composite at different magnifications. The images reveal a homogeneous distribution and dispersion of CNFs within the PEEK without the presence of CNF agglomerates. Moreover, most of CNFs do not protrude far above the poly(ether ether ketone) fracture surface, which is an indication that the interfacial bonding between CNFs and the PEEK appears to be strong [26].

### 3.2. FTIR and DSC Analysis

Figure 4 presents the FTIR spectrum at room temperature of the neat PEEK and the PEEK filled with 10 wt.% CNFs in the 600–1800 cm^−1^ range. The PEEK spectra show a carbonyl stretching vibration at 1645 cm^−1^ and skeletal ring vibrations at 1590 cm^−1^, 1487 and 1410 cm^−1^. The bending motion of C−C and (=O)−C groups appeared at around 1305 cm^−1^. The asymmetric stretching vibration bands of the diphenyl ether group appeared at 1278 cm^−1^ and 1184 cm^−1^ [27], whereas the peak at 1154 cm^−1^ corresponds to C−O−C stretching [28]. The peak at 1010 cm^−1^ is ascribed to the C–H in-plane bending vibration absorption band of the benzene ring. Absorbance at wave numbers 925 cm^−1^ to 670 cm^−1^ corresponds to C-H out of the plane bend vibrations [29]. The PEEK/10 wt.% CNF composites show a general decrease in the neat PEEK peak intensities, suggesting a decrease in PEEK chain mobility with the addition of CNFs.

The DSC analyses shown in Figure 5 were conducted to gain information about the effect of CNFs on the crystallization of PEEK, which is expected to influence the electrical properties of PEEK/10 wt.% CNF composites. In particular, the melting temperature (T_m_) and degree of crystallinity (ΔX_c_) in % of PEEK and PEEK/10 wt.% CNF composites related to the second heating scans were calculated by: (1)$${\Delta X}_{C}=\frac{{\Delta H}_{m}}{{\Delta H}_{0}f_{\mathrm{PEEK}}}\times 100\%$$

Here ΔH_m_ is the melting enthalpy of the PEEK part and ΔH_0_f_PEEK_ is the melting enthalpy of the 100% crystalline PEEK (130 J g^−1^) [30]. The corresponding values of T_m_, ΔH_m_ and ΔX_c_ are shown in Table 1. The subtle decrease in the heat flux curves in Figure 5 observed at 150 °C has to correspond to the glass transition temperature (T_g_) of PEEK [28]. This T_g_ is even more difficult to detect in PEEK/10 wt.% CNF composites. PEEK shows a clear melting peak at ~346 °C in accordance with other works [26], while PEEK/10 wt.% CNF composite shows a slightly lower melting temperature of ~345 °C. This practically unchanged T_m_ is also in agreement with previous works [26]. The results of Equation (1) show a slight decrease in ΔX_c_ from 35% corresponding to PEEK to 28% for PEEK/10 wt.% CNF composites. A similar slight reduction in ΔX_c_ was already reported for PEEK/CNF composites [26]. However, this finding is in contrast to CNFs dispersed in polypropylene (PP) [31]. In that study, the same type of CNFs (Pyrograf^®^ III PR 24 LHT XT) act as nucleation sites of PP, evidenced by a significant ΔX_c_ increase from 38% of unfilled PP to 50% in PP/CNF composites with 0.9 vol % of CNFs. 

### 3.3. Thermoelectric Properties of PEEK/CNF Composites at 30 °C

The thermoelectric properties (σ and S) at 30 °C (303.15 K) of the CNF powder and PEEK/10 wt.% CNF composites are presented in Figure 6. The CNF powder shows an σ of 133.5 ± 0.4 S m^−1^ (Table 2). Thus, the σ of the CNF powder used in this study (Pyrograf^®^ III PR 24 LHT XT) is comparable with the σ of 136.4 S m^−1^ reported for CNF powder (Pyrograf^®^ III PR 19 LHT XT) [32]. As expected, the σ of PEEK/10 wt.% CNF composites (27.5 ± 0.1 S m^−1^) is significantly less than the σ of the CNF powder. This significant difference is attributed to the wrapping of PEEK chains around the CNFs, which must increase the contact electrical resistance between the adjacent CNFs, resulting in the decrease in the CNF network conductivity [33]. It must be noted that the composite with 5 wt.% was not conductive by showing nearly the same conductivity as the pure PEEK. The σ of PEEK/7.5 wt.% CNF composite achieves a value of 5 × 10^−6^ S m^−1^ at 30 °C and therefore the PEEK/10 wt.% CNF composite analysed can be considered well above the electrical percolation threshold. In comparative terms, the σ of PEEK/10 wt.% CNF composite (27.5 S m^−1^) is higher than the values of σ achieved in melt-mixed composites of PEEK and 10 wt.% of MWCNTs [34,35]. However, a high σ of 65 S m^−1^ has been recently reported for melt-mixed composites of PEEK and 5 wt.% single-walled carbon nanotubes (SWCNTs) [17].

The Seebeck coefficients of the CNF powder and PEEK/10 wt.% CNF composite at 30 °C are also shown in Figure 6 and Table 2. The CNF powder shows n-type character, with S of −5.3 ± 0.1 μV K^−1^. It can be noticed that the S-value obtained (−5.3 μV K^−1^) is very similar to the −5.1 μV K^−1^ reported for the Pyrograf^®^ III CNF PR 19 LHT XT grade [32]. On the other hand, the S of PEEK/10 wt.% CNF composite (presented as red triangles in Figure 6) shows also a negative S-value of −3.4 ± 0.1 μV K^−1^ (Table 2). Therefore, the presence of these CNFs imparts n-type character to the PEEK/10 wt.% CNF composite, though the latter one present less negative S-values compared with as-received CNFs. In order to investigate the role of PEEK in these results, adsorbed PEEK oligomers on an idealized external graphene CNF flake were modelled as described in previous Section 2.7. The model reveals a charge transfer of approximately 0.02 eV/monomer from graphene to the PEEK oligomers in all modelled cases. This matches well with a similar finding observed on melt-mixed PP/CNF composites [14]. Furthermore, the computed total charge transfer for a comparable-length PP polymer is of a similar magnitude to what is obtained here for PEEK under similar conditions. According to this model, a slight electron withdrawing from the CNF outer layers by the PEEK molecules is expected, which would justify the lower S (absolute value) found in PEEK/CNF 10 wt.% composites. 

From this conclusion, the same type of electron withdrawing by the PEEK matrix of other carbon structures such as carbon nanotubes (CNTs) is expected, as shown in previously studied PEEK/CNT composites with p-type character [15,17]. For example, the Seebeck coefficient measured for melt-mixed composites PEEK/SWCNT composites (0.5–1.25 wt.% SWCNT of the type Tuball^TM^) [17] was higher than that of pure SWCNT powder [16]. Likewise, the values measured for PEEK composites with 3 and 5 wt.% MWCNTs of the type Nanocyl™NC 7000 [17] are also higher than those of the MWCNTs [16], and the same applies for PEEK composites with 0.5 to 3 wt.% MWCNTs of the type CNS-PEG [16,17]. In a different work, it was also found that a melt-mixed PEEK composites with 3 and 4 wt.% MWCNTs (Nanocyl™ NC 7000 type) [15] led to higher S-values than the starting MWCNT material [16]. In summary, all these findings indicate that also in PEEK composites with CNTs, the PEEK matrix induces an electron withdrawing from the CNTs, which are forming the electrically conductive network and are responsible for the Seebeck effect.

The power factor PF $(S^{2}\sigma$) at 30 °C of the PEEK/10 wt.% CNF composite and CNF powder was calculated, and the results are shown in Table 2. The CNF powder shows a PF of 3.7 × 10^−3^ μW·m^−1^·K^−2^, whereas the PEEK/10 wt.% CNF composite achieves a PF of 3.1 × 10^−4^ μW·m^−1^·K^−2^, which is slightly higher than the PF of 1.8 × 10^−4^ obtained in PP/CNF composites with 5 wt.% of Pyrograf^®^ III PR 19 LHT XT CNFs [32]. These values are one order of magnitude lower than the values of 1.0 × 10^−3^ μW·m^−1^·K^−2^ obtained in melt-mixed PEEK composites prepared with 4 wt.% MWCNTs [15]. It has to be noted that PF of 2.0 × 10^−2^ has been reported for melt-mixed composites of PEEK with 1.25 wt.% SWCNTs [17]. The conductivity, Seebeck coefficient, and power factor of all these carbon materials and their melt-processed PEEK composites are conveniently summarised in the Appendix A.

Finally, the figure of merit ($\mathrm{zT}=\frac{S^{2}\sigma }{k}\;T$) at 30 °C of the CNF powder was calculated using the thermal conductivity of 0.43 W m^−1^ K^−1^ reported on anisotropic paper-like mats of Pyrograf^®^-III CNFs [36]. The zT of PEEK/10 wt.% CNF composites was calculated from the thermal conductivity tested as described in Section 2.5. The estimated zT of the CNF powder presented a value of 2.6 × 10^−6^ (Table 2), while a lower zT of 2.3 × 10^−7^ was calculated for PEEK/CNF 10 wt.% composites, which is similar to the zT of 2.2 × 10^−7^ obtained in PP/CNF composite with 5 wt.% of Pyrograf^®^ III PR 19 LHT XT CNFs [32]. In comparative terms, the zT calculated for PEEK/CNF composites is one order of magnitude lower than the zT of ~4 × 10^−6^ at 40 °C reported for PEEK composites filled with 4 wt.% of MWCNTs and 3 wt.% of graphite nanoplates [15].

### 3.4. Thermoelectric Analysis of PEEK/CNF Composites from 30 °C to 100 °C

The thermoelectric properties σ(T) and S(T) from 30 °C (303 K) to 100 °C (373 K) of the CNF powder and PEEK/10 wt.% CNF composites are also analysed in this study (Figure 6). As was previously noted, σ of 133.5 ± 0.4 S m^−1^ is obtained for the CNF powder at 30 °C, which decreases up to 123.9 ± 14.1 S m^−1^ at 100 °C (Table 3). Thus, the σ (100 °C) value of the CNF powder used in this study (Pyrograf^®^ III PR 24 LHT XT) is comparable to the σ (100 °C) of 127 S m^−1^ reported for CNF powder (Pyrograf^®^ III PR 19 LHT XT) [32]. Interestingly, the CNF powder shows a slight positive temperature effect (dσ/dT < 0) as it was observed for Pyrograf^®^ III PR 19 LHT XT CNF powder over the same interval of temperatures [32]. In this regard, it is noted that CNTs usually present a negative temperature effect (dσ/dT > 0) [37,38,39]. In contrast to the σ (T) of the CNF powder, the PEEK/10 wt.% CNF composite shows a very slight negative temperature effect from ~27.5 ± 0.1 S m^−1^ at 30 °C to ~27.9 ± 0.1 S m^−1^ at 100 °C (black triangle icons in Figure 6), which is consistent with results in other articles [40,41]. Therefore, the σ (T) of PEEK/10 wt.% CNF composite does not follow the same σ (T) found in the CNF powder. In other words, the PEEK matrix should play its role on the σ (T) of PEEK/CNF composites. This finding is in contrast to previous work [32], where the σ (T) of both the PP/5 wt.% CNF composite and the corresponding CNFs showed a positive temperature effect.

The S (T) of the CNF powder is also presented as red circle icons in Figure 6. The n-type character of the CNF powder found at 30 °C keeps negative at all temperatures. In particular, the CNF powder presents a S-value of −5.3 ± 0.1 μV K^−1^ at 30 °C, which increases gradually (in absolute value) up to −5.9 ± 0.1 μV K^−1^ at 100 °C (Table 3). It must be noticed that the S-value obtained here at 100 °C is very similar to the value of −5.8 μV K^−1^ reported for the Pyrograf^®^ III CNF PR 19 LHT XT grade at 100 °C [32]. Based on the experimental values presented here, it can be concluded that both Pyrograf^®^ III grades (PR 19 LHT XT and PR 24 LHT XT) present similar σ (T) and S (T) in the interval of 30−100 °C. On the other hand, the S (T) of PEEK/10 wt.% CNF composite (red triangle icons in Figure 6) shows S-values from −3.4 ± 0.1 μV K^−1^ at 30 °C to −4.4 ± 0.1 μV K^−1^ at 100 °C (Table 3). Therefore, the S (T) of the PEEK/10 wt.% CNF composite, similarly to the S (T) of the CNF powder, shows negative S-values gradually increasing (in absolute value) with temperature. 

The power factor PF as function of temperature of the PEEK/10 wt.% CNF composite and CNF powder was calculated, and the results are shown in Table 3. At 100 °C, the CNF powder achieves a value of 4.3 × 10^−3^ μW·m^−1^·K^−2^, whereas the PEEK/10 wt.% CNF composite achieves a PF of 5.5 × 10^−4^ μW·m^−1^·K^−2^. Thus, this PF is slightly higher than the PF of 2.6 × 10^−4^ obtained at 100 °C in PP/CNF composite with 5 wt.% of Pyrograf^®^ III PR 19 LHT XT CNFs [32]. The figure of merit zT of PEEK/10 wt.% CNF composite increases slightly from 2.3 × 10^−7^ (30 °C) up to 4.7 × 10^−7^ (100 °C) despite the slight increase in the thermal conductivity with temperature (Table 3). A similar zT of 4.4 × 10^−7^ at 100 °C was found in PP/CNF composite with 5 wt.% of Pyrograf^®^ III PR 19 LHT XT CNFs [32].

### 3.5. Electronic Modelling of PEEK/CNF Composites

Similar to the previous work [32], the 3D variable-range hopping (VRH) model is applied to evaluate the σ (T) of the CNF powder and PEEK/10 wt.% CNF composite [42,43]: (2)$$\sigma \left(T\right)=\sigma_{0\;}\;\exp[\pm (\frac{T_{C}}{T}){\;}^{\frac{1}{4}}\;]$$

Here σ_0_ is the conductivity at an infinite temperature, and $T_{C}\equiv \;\frac{\left|W_{D}\right|}{k_{B}}$ is a characteristic temperature scale determined by the average energy potential barrier (W_D_ < 0) or potential well (W_D_ > 0), respectively, and k_B_ is the Boltzmann’s constant. It is important to notice that when W_D_ > 0, Equation (2) describes a thermally activated hopping mechanism across a random network of potential wells, leading to a typical dσ/dT > 0, while when W_D_ < 0, Equation (2) describes a thermally activated scattering mechanism across a random distribution of impurities or structural defects, leading to a typical dσ/dT < 0. Values of$\;\sigma_{0}=27.60{\;S\;m}^{-1}$, $T_{C}=1.87\times 10^{3}\;K$, and $W_{D}=-160\;\mathrm{meV}$ as shown in Table 4 are calculated from Equation (2) for the CNF powder. Thus, it is deduced that the parameters σ_0_, T_C_ and W_D_ of this Pyrograf^®^ III grade (PR 24 LHT XT) are comparable with the σ_0_, T_C_ and W_D_ calculated by the VRH model for Pyrograf^®^ III PR 19 LHT XT [32]. Interestingly, T_C_ (1.9 × 10^3^ K) is one order of magnitude higher than that of some reported SWCNT mats (2.5 × 10^2^ K) [44]. The W_D_ (absolute value) for the CNF powder (160 meV) is one order of magnitude higher than the activation energy (60 meV) reported for n-type graphitized carbon fibres in the 250–750 K interval [45]. Similar to the previous work [32], the CNF powder used in this study shows negative W_D_. This finding, which is not usual in carbon nanostructures [43,46,47], can be explained by the presence of impurities, such as the oxygen detected by X-ray photoelectron spectroscopy (XPS) in the same type of Pyrograf^®^ III CNFs [23,48,49]. These impurities could activate a thermal-enhanced backscattering mechanism due to the presence of virtual bound-states, represented as sharp peaks near the E_F_ in the density of states [46,50].

The σ (T) of PEEK/10 wt.% CNF composites was also studied with the 3D VRH model, from which$\;\sigma_{0}=33.8\;{S\;m}^{-1}$,$\;T_{C}=0.5\;K$, and $W_{D}=+4.3\times 10^{-2}\;\mathrm{meV}$ were obtained. Thus, as can be seen in Table 4, the T_C_ obtained for the PEEK/10 wt.% CNF composite is very different (four orders of magnitude lower) than the T_C_ (1.9 × 10^3^ K) of the CNF powder. Notably, the W_D_ of PEEK/10 wt.% CNF composite is now positive. These results imply that the σ (T) of PEEK/10 wt.% CNF composite is not dominated by the thermally activated backscattering mechanism responsible for the negative W_D_ found in the CNF powder. Thus, the σ (T) of PEEK/10 wt.% CNF composite can be understood as the e^−^ overcoming by hopping in a random network of potential wells [43,49,51].

The S (T) of the CNF powder and PEEK/10 wt.% CNF composite is depicted by the same model proposed for describing the nonlinear S(T) of polypropylene composites melt-processed with similar CNFs (PR 19 LHT XT) [32]: (3)$$S\;\left(T\right)=\mathrm{bT}+\frac{{\mathrm{cT}}_{p}}{T^{2}}\frac{\exp\left(\frac{T_{P}}{T}\right)}{{\left[\exp\left(\frac{T_{P}}{T}\right)+1\right]}^{2}}$$

Here, bT represents the metallic (linear) term, c is a constant, and $T_{p}=\left(E_{p}-E_{F}\right)/k$ where k_B_ is Boltzmann’s constant, E_F_ is the Fermi energy level, and E_P_ is the energy corresponding to the sharply varying and localized states near E_F_ in the density of states due to the contribution of impurities [46,50]. The best fit of S(T) with Equation (3) for the CNF powder (Figure 6) shows that the first term b is positive with $5.1\times 10^{-3}\;{\mu V\;K}^{-2}$, while the second term c is negative with $-1.8\times 10^{4}\;\mu V\;$and $T_{p}=997.42\;K$, yielding a $E_{P\;}-E_{F}=0.086\;\mathrm{eV}$. The negative sign of c can be physically interpreted as the resonances near the E_F_ at the density of states caused by impurities present in the CNF structure [46]. As expected from the experimental S(T) values (Table 3), the b, c, T_P_ and $E_{P\;}-E_{F}\;$values obtained by Equation (3) for this Pyrograf^®^ III grade (PR 24 LHT XT) are very similar to the b, c, T_P_ and $E_{P\;}-E_{F}\;$values calculated by the same model for Pyrograf^®^ III PR 19 LHT XT [32] (Table 5). Likewise, the S (T) of PEEK/10 wt.% CNF composite is also fitted by Equation (3) with the best fit resulting in $b=1.85\times 10^{-4}\;\mu V\;K^{-2}$, $c=-1.25\times 10^{4}\;\mu V$, $T_{p}=1117.6\;K$, and $E_{P\;}-E_{F}=0.096\;\mathrm{eV}$. As shown in Table 5, the fittings obtained by Equation (3) for PEEK/10 wt.% CNF are very similar to parameters calculated for the CNF powder. Thus, it can be deduced that in contrast to σ (T), the S (T) of CNF powder clearly drives the S (T) of PEEK/10 wt.% CNF composite.

## 4. Conclusions

The electrical conductivity (σ) and Seebeck coefficient (S) between 30 °C and 100 °C of as-received carbon nanofibers (CNFs) and melt-processed PEEK/10 wt.% CNF composites are analysed. At 30 °C, the σ, S, and power factor (PF) of the CNFs are ~133 S m^−1^, −5.3 μV K^−1^, and 3.7 × 10^−3^ μWm^−1^ K^−2^, respectively. The PEEK/10 wt.% CNF composite shows lower conductivity and S of 27.5 S m^−1^ and −3.4 μV K^−1^, respectively, corresponding to a PF of 3.1 × 10^−4^ μWm^−1^ K^−2^. As the CNFs, the prepared PEEK/CNF composites are n-type materials with e^−^ as dominant charge carriers, and the origin of their n-type character is explained by the presence of impurities in the CNFs, which could produce sharp peaks close to the Fermi energy level (E_F_) of CNFs. The less negative value of S in the PEEK/CNF 10 wt.% composite when compared with the as-received CNFs is rationalized with the help of quantum chemical computer models and attributed to an electron-withdrawing effect arising from the PEEK molecules in contact with the most external graphene layers of CNFs. Moreover, in contrast to the slight positive temperature effect of σ found in the as-received CNFs, the σ (T) of PEEK/CNF 10 wt.% composite from 30 °C to 100 °C shows a very slight negative temperature effect. Thus, it is concluded that the σ (T) of the latter does not follow the same σ (T) found in CNF powder, and thereby the presence of PEEK must play a predominant role in their mechanism conduction. This fact is better understood through the 3D variable-range hopping model of the σ (T) in PEEK/CNF 10 wt.% composite, which points towards the e^−^ overcoming in a random network of potential wells by hopping. In addition, the S (T) of PEEK/CNF 10 wt.% composite from 30 °C to 100 °C presents a negative temperature effect, as does the S (T) of the CNFs for the same range of temperature, which was analysed using the same model proposed for describing the nonlinear S(T) of melt-processed polypropylene/CNF composites.

## Figures and Tables

**Figure 1 polymers-14-04803-f001:**
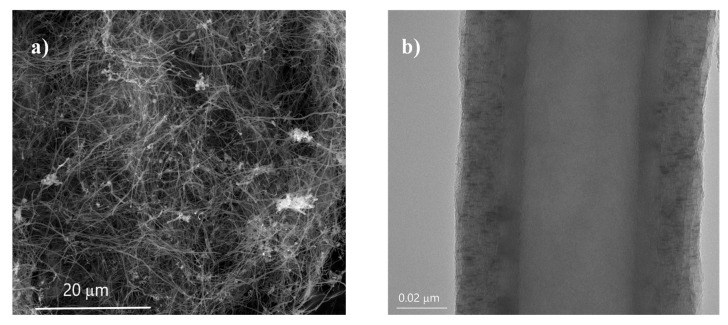
(**a**) SEM image of as-received carbon nanofiber powder, (**b**) TEM image of single CNF (Pyrograf^®^ III PR 24 LHT XT).

**Figure 2 polymers-14-04803-f002:**
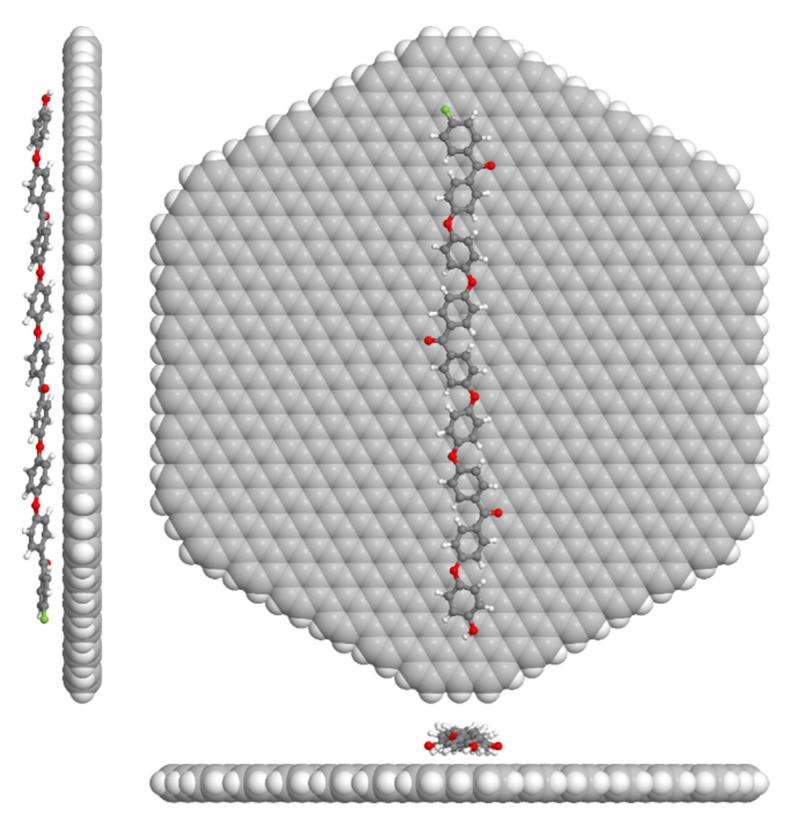
Computer model of PEEK adsorbed on a C_834_H_78_ graphene flake.

**Figure 3 polymers-14-04803-f003:**
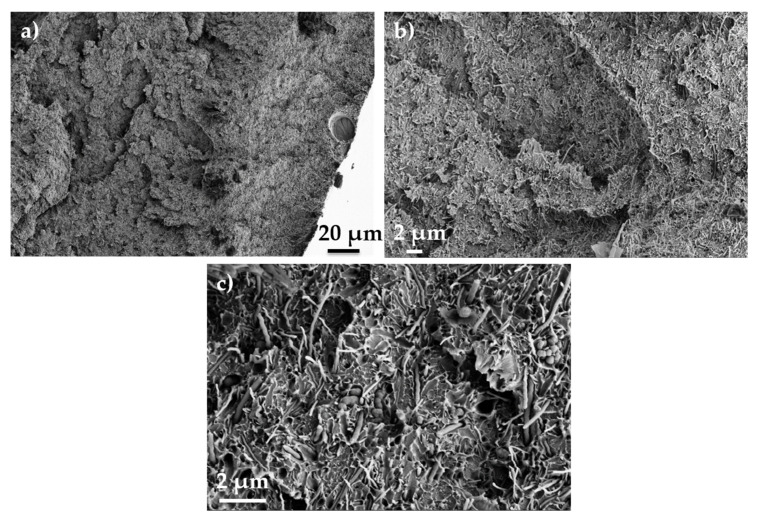
SEM micrographs of the PEEK/10 wt.% CNF composite at (**a**) lower, and (**b**,**c**) higher magnifications.

**Figure 4 polymers-14-04803-f004:**
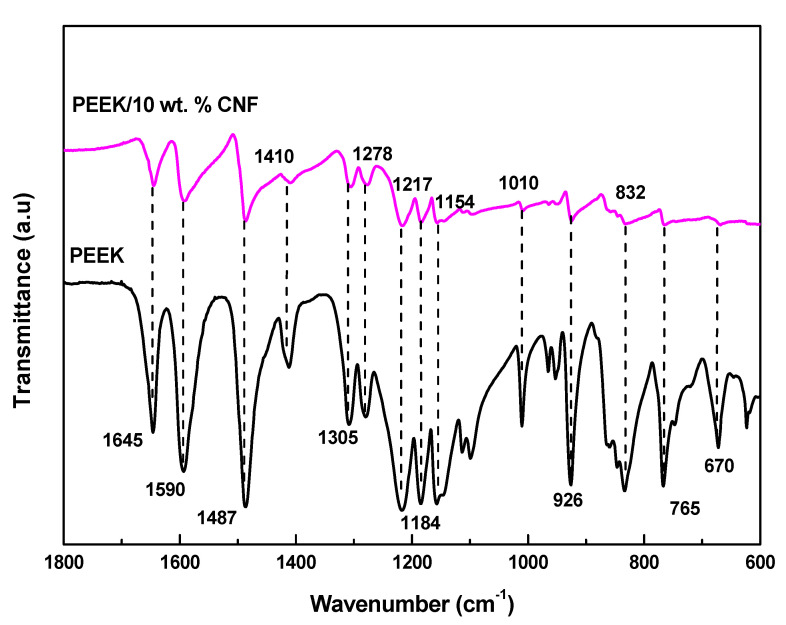
FTIR of PEEK and PEEK/10 wt.% CNF composites (short dot lines are to guide the eyes).

**Figure 5 polymers-14-04803-f005:**
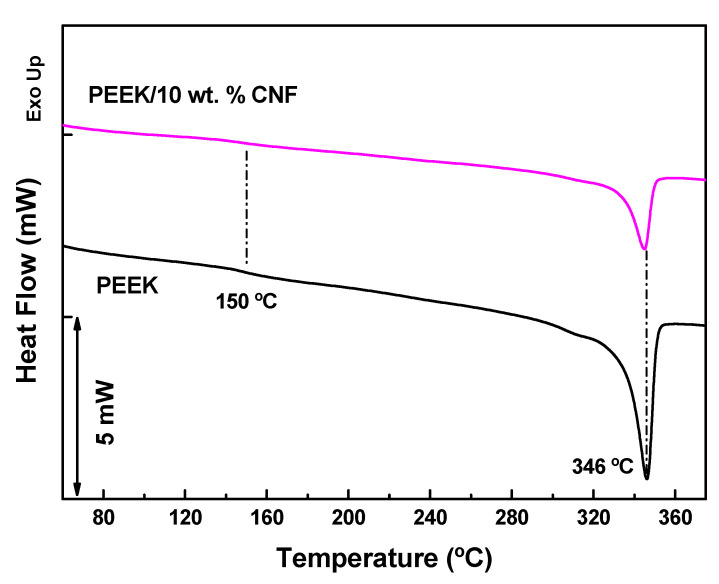
DSC thermographs of PEEK and PEEK/10 wt.% CNF composites (dotted lines are guides for the eyes).

**Figure 6 polymers-14-04803-f006:**
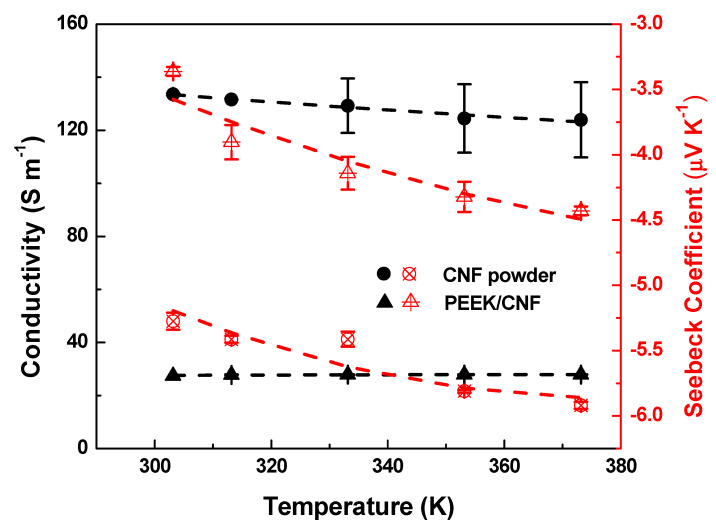
Electrical volume conductivity (black icons) and Seebeck coefficient (red icons) of CNF powder (circle icons) and PEEK/10 wt.% CNF composite (triangle icons). The black and red dash lines represent the fitting with Equations (2) and (3), respectively.

**Table 1 polymers-14-04803-t001:** DSC data of neat PEEK and PEEK/10 wt.% CNF composites corresponding to the second heating scans.

Sample	T_m_ (°C)	ΔH_m_ (J g^−1^)	ΔX_c_ (%)
PEEK	346.0	46.2	35.5
PEEK/10 wt.% CNF	345.2	34.6	28.0

**Table 2 polymers-14-04803-t002:** Electrical conductivity σ, Seebeck coefficient S, power factor PF, and figure of merit zT of PEEK/10 wt.% CNF composites and CNF powder at 30 °C.

Sample	σ	S ( μV·K^−1^)	PF (μW·m^−1^·K^−2^)	zT
PEEK/10 wt.% CNF	27.5 ± 0.1	−3.4 ± 0.03	3.1 ± 0.1 × 10^−3^	2.3 × 10^−7^
CNF power	133.5 ± 0.4	−5.3 ± 0.08	3.7 ± 0.1× 10^−3^	2.6 × 10^−6^

**Table 3 polymers-14-04803-t003:** Thermoelectric properties of CNF powder and PEEK/10 wt.% CNF composite.

	CNF Powder			PEEK/10 wt.% CNF				
T (°C)	σ (S m^−1^)	S (µV K^−1^)	P F (μW m^−1^ K^−2^)	σ (S m^−1^)	S (µV K^−1^)	P F (μW m^−1^ K^−2^)	k (W m^−1^ K^−1^)	zT
30	133.5 ± 0.4	−5.3 ± 0.1	3.7 × 10^−3^	27.5 ± 0.1	−3.4 ± 0.1	3.1 × 10^−4^	0.41	2.3 × 10^−7^
40	131.6 ± 0.1	−5.4 ± 0.1	3.8 × 10^−3^	27.8 ± 0.1	−3.9 ± 0.1	4.2 × 10^−4^	0.42	3.2 × 10^−7^
60	129.2 ± 10.3	−5.4 ± 0.1	3.8 × 10^−3^	27.8 ± 0.1	−4.1 ± 0.1	4.8 × 10^−4^	0.44	3.6 × 10^−7^
80	124.4 ± 12.9	−5.8 ± 0.1	4.2 × 10^−3^	27.8 ± 0.1	−4.3 ± 0.1	5.2 × 10^−4^	0.45	4.1 × 10^−7^
100	123.9 ± 14.1	−5.9 ± 0.1	4.3 × 10^−3^	27.9 ± 0.1	−4.4 ± 0.1	5.5 × 10^−4^	0.43	4.7 × 10^−7^

**Table 4 polymers-14-04803-t004:** Parameters σ_0_, T_C_, and W_D_ extracted from VRH model of CNF powder and PP/5 wt.% CNF composites analysed in previous study [32], and CNF powder and PEEK/10 wt.% CNF composites analysed in this study.

CNF Grade	Polymer	Method	CNF Content	σ_0_ (S m^−1^)	T_C_ (K)	W_D_ (meV)
PR 19 LHT XT	−	−	100 wt.% (powder)	46.40	3.9 × 10^2^	−34
	PP	Melt-mixing	5 wt.%	1.76	7.4 × 10^3^	−640
PR 24 LHT XT	−	−	100 wt.% (powder)	27.60	1.9 × 10^3^	−160
	PEEK	Melt-mixing	10 wt.%	33.80	5 × 10^−1^	4.3 × 10^−2^

**Table 5 polymers-14-04803-t005:** Parameters b, c, T_P_ and $E_{P\;}-E_{F}\;$ obtained by Equation (3) of CNF powder and PP/5 wt.% CNF composites analysed in a previous study [32], and CNF powder and PEEK/10 wt.% CNF composites analysed in this study.

CNF Grade	Polymer	Method	CNF Content	b (μV K^−2^)	c (μV)	T_p_ (K)	E_p_ − E_F_ (eV)
PR 19 LHT XT	-	-	100 wt.% (powder)	5.5 × 10^−3^	−1.8 × 10^4^	9.9 × 10^2^	8.6 × 10^−2^
	PP	Melt-mixing	5 wt.%	1.5 × 10^−3^	−1.3 × 10^4^	1.1 × 10^3^	9.4 × 10^−2^
PR 24 LHT XT	-	-	100 wt.% (powder)	5.1 × 10^−3^	−1.8 × 10^4^	9.9 × 10^2^	8.6 × 10^−2^
	PEEK	Melt-mixing	10 wt.%	1.8 × 10^−4^	−1.2 × 10^4^	1.1 × 10^3^	9.6 × 10^−2^

## Data Availability

The authors confirm that the data supporting the findings of this study are available within the article.

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
