# Peer review of "Thermoelectric Properties of N-Type Poly (Ether Ether Ketone)/Carbon Nanofiber Melt-Processed Composites"

_polymers, 2022, doi:10.3390/polym14224803_

Round 1

Reviewer 1 Report

1. Section 2.7, Figure 2 should be changed or deleted. This paper discussed CNF rather than xTB-GFN1. GFN should be defined.

2. Section 3.1, The mean diameter of CNF should be obtained from your experimental measurement.  Why references [14, 20] cited?

3. Section 3.1, "without the presence of CNF agglomerates".  No data can support this point.

4. I think more characterization for the CNF/PEEK (TEM, XPS) should be added to improve the quality of the paper.

5. I have no doubt the authors are expert in this area, but I struggle to understand the motivation for this work. 

Reviewer 2 Report

Authors have shown that the developed composite based PEEK and CNF and their electrical properties. Overall this is a comprehensive manuscript and this work will be in the field of interest of researcher’s community, for that I recommended the publication of this paper after these minor corrections.

1/ The introduction is well long it should be reduced specially in the beginning? You should also developed more the aim and the objectify the application of the present work ?

2/ the discussion of Figure 1 (specially TEM micrograph) was not deeply described, in addition you should improve the quality of figure.

3/ Authors should give the Raman spectra of the composite compared to the CNF and PEEK?

4/ Are the results reproducible?

Round 2

Reviewer 1 Report

This paper can be published.